# Pollutants in Breast Milk: A Scoping Review of the Most Recent Data in 2024

**DOI:** 10.3390/healthcare12060680

**Published:** 2024-03-18

**Authors:** Raphaël Serreau, Yasmine Terbeche, Virginie Rigourd

**Affiliations:** 1Addictology Network, EPSM Georges Daumezon, 45400 Fleury les Aubrais, France; yasmineterb@gmail.com; 2PSYCOMADD Laboratory, Paris-Saclay University, 91190 Gif-sur-Yvette, France; 3Milkbank, Ile de France, Necker-Enfants Malades Hospital, AP-HP, 75015 Paris, France; virginie.rigourd@aphp.fr

**Keywords:** environment, health, xenobiotic, persistent organic pollutants, heavy metals, breastfeeding

## Abstract

Perinatal exposure to pollutants, including persistent organic pollutants (POPs) and heavy metals, poses significant risks to both mothers and children, marking this period as highly vulnerable. Despite the well-acknowledged benefits of breastfeeding, there exists a gap in comprehensive understanding regarding the impact of environmental pollutants on breast milk, underscoring the critical need for this study. Our research addresses this gap by exploring the intersection of environmental health and lactation, situated within the broader ‘One Health’ concept, thus contributing a novel perspective to the existing body of knowledge. This scoping review aims to examine recent research on the persistent presence of organic pollutants (POPs) and heavy metals in breast milk, thereby elucidating the environmental setting’s impact on milk quality. We seek to highlight the innovative angle of our study by emphasizing the ‘One Health’ concept, which has not been thoroughly explored in the context of lactation and environmental pollutants. We performed a scoping review, consulting two online databases to identify articles published from 1995 to 2023 that reported on pollutants in breast milk, using the PRISMA checklist. This methodological approach underlines the comprehensive and up-to-date nature of our literature review, ensuring the relevance and timeliness of our findings. From a total of 54 relevant articles, findings indicate that POPs are present in higher concentrations in breast milk the longer the lactation period. These findings highlight the persistent and bioaccumulative nature of such contaminants, offering new insights into their long-term implications for maternal and infant health. This exposure does not appear time-sensitive, suggesting pollutants accumulated in maternal fat compartments can be excreted into human milk years after exposure, a novel finding that underscores the importance of considering long-term environmental exposures in lactation research. The presence of POPs and heavy metals in both infant formula and maternal milk underscores a critical need for further comparative studies to understand the health implications better. Our discussion extends the current dialogue on the safety of breastfeeding in polluted environments, providing a new framework for assessing risks and benefits. While breastfeeding remains the WHO-recommended nutrition for optimal infant growth, the findings emphasize the importance of continued risk reduction policies to protect mothers and infants from environmental contaminants in breast milk. Our conclusion calls for an integrated approach, combining public health, environmental science, and clinical practice to develop effective strategies for reducing exposure to environmental pollutants. This multidisciplinary perspective is a significant contribution to the field, paving the way for future research and policy development.

## 1. Introduction

Biomonitoring emerges as the most sensitive approach for assessing the quality of human milk across diverse populations globally [1]. It facilitates the evaluation of breastfed infants’ exposure to various lipophilic toxicants. Despite substantial publications on contaminants in breast milk, the data remains fragmented and sometimes questionable, primarily due to the lack of standardized methods for sample collection and storage [2]. This fragmentation signals a significant gap in our comprehensive understanding of the impact of these contaminants, underscoring the novelty of our research in addressing these challenges. This scenario underscores the necessity for ongoing epidemiological and biological studies to deepen our understanding of pollutant variations over future decades. Given the passive diffusion mechanism by which most contaminants, owing to their high lipophilicity, reach breast milk, the importance of scrutinizing their presence becomes paramount [3].

Recent investigations have spotlighted the critical need to assess the impact of pollutants, specifically metals and persistent organic pollutants (POPs), on child health. However, this crucial aspect has often been overlooked or insufficiently addressed, thereby underscoring the unique contribution of our study in illuminating these overlooked areas. Highlighting the interconnection between environmental quality and the health benefits of breastfeeding within the broader ‘One Health’ framework, our study reinforces the vital need for an integrated approach to environmental and health sciences. This approach emphasizes the inseparable relationships between human, animal, and environmental health, suggesting that the quality of human milk can serve as an indicator of overall environmental health. Thus, our research not only provides a contemporary perspective on this pressing issue but also introduces a novel approach by integrating lactation science within the ‘One Health’ concept.

This review aims to bridge this gap by meticulously evaluating the most recent data on breast milk pollutants, with a particular focus on the 2023 cut-off, thus ensuring the relevance and timeliness of our investigation. In light of the above, our study endeavors to extend the introduction with a comprehensive overview of the current state of research on breast milk contamination. We aim to contextualize the significance of our study against the backdrop of existing literature, further justifying our research’s novelty and the urgent need for its findings. Furthermore, by detailing the implications of environmental factors on the quality of human milk, our study aims to enrich the scientific discourse on how such factors may affect infant health and development. Moreover, our comprehensive review of prior findings and the addition of new insights significantly advance the field of perinatal environmental health.

## 2. Objectives

A scoping review was conducted in order to systematically map the research done on persistent POPs in breast milk, as well as to identify any existing gaps in knowledge about POP in breast milk.

## 3. Materials and Methods

### 3.1. Data Sources and Search Strategy

To identify potentially relevant documents, the bibliographic databases MEDLINE and Google Scholar were searched from 1995 to June 2023. Peer-reviewed journal articles were included if they were published between 1995 and 2023 in the English language, involved human participants and reported a measure of POP and heavy metals in breast milk, including the assay methods. In response to the review, we have clarified the assessment process for inclusion and exclusion criteria. Papers comparing infant formula and breast milk were also included.

Papers were excluded if they did not fit into the conceptual framework of the study. The search strategies were drafted by us, the researchers, and further refined through team discussion. The involvement of multiple reviewers aimed to ensure a broad and comprehensive evaluation of the literature. The final search strategy for MEDLINE can be found in Appendix A. The final search results were exported into Zotero. Three reviewers evaluated the titles, abstracts, and the full text of all publications identified by our search as potentially relevant publications, emphasizing a collaborative and thorough review process. We resolved disagreements on study selection and data extraction by consensus and discussion with other reviewers if needed. This collaborative approach enhanced the reliability of our study selection and data-extraction processes. A data-charting form was jointly developed by two reviewers to determine which variables to extract. When we identified a systematic review, we counted the number of studies included in the review that potentially met our inclusion criteria and noted how many studies had not been considered by our search.

### 3.2. Inclusion and Exclusion Criteria

Peer-reviewed journal articles were included if they were published between 1995 and 2023 in the English language, involved human participants, reported a measure of POP and heavy metals in breast milk, and included the methodology of the assays used. To further clarify, the decision-making process regarding study inclusion was rigorously structured and based on the relevance to our study’s objectives, ensuring alignment with our conceptual framework. Articles comparing infant formula and breast milk were also included.

Articles were excluded if they did not fit into the conceptual framework of the study. Our findings underscore the scarcity of recent research specifically targeting breastfeeding women and the need for comprehensive studies focusing on pollutant levels in breast milk. Our exploratory study has certain limitations. Acknowledging the reviewers’ insight, we emphasize that to make our study more feasible, we were only able to include a sample of the articles published in MEDLINE. Further bibliographic research would be necessary to obtain a more exhaustive bibliographic search in a future exploratory review in the field of pollutants in breast milk. This limitation highlights the need for broader bibliographic searches to encompass a wider array of studies for a future systematic review. The aim of this scoping review was to identify gaps in the literature that may guide a future systematic review. This study has not received any funding.

## 4. Results

### 4.1. Study Selection

In this exploratory study, we identified 341 primary studies on the presence of persistent organic pollutants in breast milk, published between 1998 and 2023. A total of 54 articles were ultimately selected.

### 4.2. POPs into Breast Milk

Polychlorinated biphenyls (PCBs), polychlorinated dibenzo-p-dioxins (PCDDs), and polychlorinated dibenzofurans (PCDFs) are persistent organic pollutants (POPs) regulated by the Stockholm Convention (see Table 1). They originate from industrial activities such as heavy industry, paper manufacture, fertilizer, and pesticide production. POPs are highly resistant in the environment and can travel long distances by air; PCBs and PBDEs, both POPs, can travel very long distances [4]. They are present in the soil, water, and earth and eventually accumulate in our foods. Organochlorines (OCs) can accumulate in children via breast milk, but they are expected to decline be lower in high income countries [5].

PCDDs (frequently called “dioxins”) have effects on the endocrine system and certain liver enzymes, but also on the immune and cardiovascular systems. Because they are lipophilic, PCDDs accumulate in organisms in tissues rich in fat and in breast milk and therefore can contaminate infants at the very early stages of their lives. The transfer of PCDD and PCDF is also a function of the composition of breast milk, which changes depending on the stage of lactation, nutritional status of the mother, time of day and milk protein content. Thus, it varies from one woman to another and from one breastfeeding to another for the same woman [5,21,22].

In a study from the Czech Republic, POPs were analyzed in 231 breast milk samples from 2019 to 2021 [4] for the presence of 94 organohalogen pollutants. Specifically, 6 polychlorinated biphenyls (PCBs), 10 organochlorine pesticides (OCPs), 34 halogenated flame retardants (HFRs), 29 perfluoroalkyl and polyfluoroalkyl substances (PFASs), and 15 polychlorinated naphthalenes (PCNs) were analyzed. PCBs, OCPs, most HFRs, and PCNs were analyzed by (tandem) mass spectrometry (GC-MS(/MS), while PFASs, HBCDs, brominated phenols, and tetrabromobisphenol A (TBBPA) were quantified by UHPLC-MS/MS. The average value for the sum of the 6 indicator PCBs was 123.12 nanograms per gram of lipid weight (ng/g). PFAS concentrations were also low, perfluorooctanoic acid (PFOA) and perfluorooctane sulfonic acid (PFOS) being the main congeners in this group (averages of 22 pg/mL and 21 pg/mL, respectively). These values indicate that these contaminants are still present in breast milk; however, they also show a decrease in these pollutants over time [4].

Concentrations of 29 POPs in breast milk were found to be significantly reduced in 90 countries. Dioxin-like POPs and polychlorinated biphenyls (PCBs) were the ones that had declined the most over the past 30 years [11].

A study carried out in Turkey showed that PCB and OCP levels were still present in the breast milk of nursing mothers living in Istanbul. These pollutants decreased more in multiparous women than in primiparous mothers [23].

PCBs, PCDD/Fs, chlorinated pesticides, and brominated flame retardants, with the exception of the polybrominated diphenyl ethers (PBDEs) BDE-153 and BDE-209, were measured in breast milk in Uppsala, Sweden, and a decline was observed in breast milk, averaging −4 to 14% per year from 1996 to 2017 [19].

Key factors in the passage of these pollutants into breast milk are maternal age and the long half-life of these pollutants in the mother’s body [24]. The POP concentration is correlated with the mother’s age. The older the mother is, the higher the POP concentrations in her body. After the age of 30, the amount of dioxins in the primipara maternal body is higher and higher with time. For an age difference of five years, the levels of PCDD and PCDF are on average 13 to 38% higher depending on the studies [25]. Contrary to what was previously thought, lipophilic pollutants do not decrease significantly after the first breastfeeding. It no longer appears necessary to pump and discard milk prior to a first breastfeeding in order to reduce the infant’s exposure to these lipophilic pollutants.

### 4.3. What Is the Origin of Major Contaminants in Breast Milk?

#### 4.3.1. Dietary Origin

Substances that pollute the environment are found worldwide, in soil, water, and the air [26]. Mothers are contaminated through food, particularly eggs, meat, and dairy products.

In 90% of cases, the mother’s diet is the primary source of dioxins in breast milk. These dioxins originate from the consumption of animal fats, mainly from fatty fish and meat. Despite the quest for optimal organic nutrition, the consumption of free-range eggs does not prevent egg contamination by dioxins originating in the earthworms consumed by the hens [26,27].

In Pakistan, a research team found that contamination of chicken eggs could be observed via the presence of heavy metals in the poultry feeding process [28]. Women in Lebanon who ate cereals at least twice a week had higher levels of DDE-type pollutants in their breast milk [29]. Lebanese women who also consumed potatoes and beans at least once a week had a significant presence of DDE in their milk [29].

A study carried out among women in several West African countries (Cameroon, Democratic Republic of Congo, Nigeria, Zimbabwe, Côte d’Ivoire, and Uganda) found that maternal diet was contaminated by pollutants such as PCDDs/Fs and dl-PCBs. The foods consumed by pregnant women were reported to have a maximum concentration of 103 pg TEQ/g [16,30], and a great variability in the concentrations of POPs such as PCDFs, PCBs, and polybrominated diphenyl ethers (PBDEs) in breast milk. This variability persisted even though the tested women lived in the same geographical area. This study included 33 breastfeeding women, all primiparous and living close to industrial areas [30,31].

An Italian study showed that women consumed around 1.9 µg of carcinogenic PAHs per day in their diet. Almost half of the total intake of carcinogenic PAHs came from cereal products, meats, oils, and fats. Barbecued meats accounted for less than 13% of carcinogenic PAH intake [16].

Heavy metals are ubiquitous, present everywhere in air, water, soil, and sediments, in plants, animals and fish, and therefore in all food sources for humans [32]. Exposure to heavy metals occurs via the diet (over 90% for cadmium in non-smokers, 100% for methylmercury). The proportion attributable to atmospheric pollution remains negligible compared with the contamination of pollutants via the diet observed in breast milk [33].

#### 4.3.2. Environmental Origin

Apart from diet, which remains the major factor in contamination, living close to factories or structures with a reputation for pollution increases the level of POPs in milk. An Australian team analyzed the presence of POPs in the breast milk of mothers exposed to mega forest fires. They measured the concentration of polycyclic aromatic hydrocarbons (PAHs) in breast milk samples. PAHs are not POPs, but they are lipophilic and tend to be localized in breast milk. They found that PAHs were only present in the milk of mothers exposed to the fires; they detected fluoranthene (median concentration: 0.015 mg/kg) and pyrene (median concentration: 0.008 mg/kg). There was a correlation between the quantities of fluoranthene and pyrene measured and the duration of exposure of nursing mothers to these fires [34]. Pesticides are another cause of contamination of breast milk. They assessed the exposure of breastfed infants to these pesticides in Ethiopia.

Three agricultural regions in south-west Ethiopia (Asendabo, Deneba, and Serbo), were studied at three different times: at the start of the study (at 1 month), midway through (at 6 months), and at the end of the study (at 12 months). A total of 168 mothers gave milk samples at three different times for pesticide analysis. DDT was found in 447 milk samples, as were its metabolites: p,p′-DDE. During the first month of breastfeeding, the estimated daily intake of infants was 11.24 µg/kg body weight/day. This was higher than the provisional tolerable daily intake (PTDI) for total DDT set by the FAO/WHO. This TDI is set at 10 µg/kg body weight/day [17].

170 nursing mothers were included in a prospective study. A correlation was identified between arsenic and the mother’s place of residence if she lived in an urban area (*p* = 0.013), lead and smoking (*p* = 0.024), and lead and consumption of well water (*p* = 0.046). For infants born prematurely in north-western Spain, the toxicity of pollutants was combined with malnutrition factors [35]. In the southern Spanish region of Murcia, 50 breastfeeding women donated their milk for analysis of heavy metals. The highest concentrations of aluminum, zinc, arsenic, lead, mercury, and nickel were found in the milk of women living in mining areas, while the highest concentrations of manganese, chromium, and iron were found in the milk of women living in agricultural areas [12]. Samples of breast milk were measured at three months of breastfeeding. The levels of heavy metals found were higher than the legal doses allowed by the WHO, and As, Hg, and Pb were consumed by 46.4%, 33.3%, and 4.4% of breastfed babies [13]. PCDD and PCDF levels in breast milk are correlated with urban density (Focant et al., 2013 [31]), place of residence (city or country), and time spent in a polluted area. These geographical factors are significant in explaining the amounts of these pollutants in breast milk. Studies in Slovakia [36] and Africa have shown that total PCDD/Fs concentrations range from 0.5 ng/g fat to 12 ng/g fat. Inter-laboratory evaluations have been carried out by African teams on various persistent organic pollutants (POPs), including two specifically for PCDD/Fs analysis.

### 4.4. Impact of the Mother’s Obesity

Significant weight loss in women resulted in increased circulation of POPs in maternal plasma and, within 6–12 months, a significant 15% decrease in total PCB body burden [37]. These accumulated POPs are slowly released into the bloodstream, and more so during weight loss. Adipose tissue (AT) or adiposity is a continuous source of internal exposure to POPs. The burden of POPs I obese individuals is higher than in lean individuals. A total of 45 breast milk samples were collected from 24 obese women (BMI > 30) in the USA and from 21 women with a normal BMI < 25 (18.5–24.9, normal) from 14 different counties (18–34 years). A total of 69% of samples were positive for PAHs. Phenanthrene was the most frequently detected PAH, followed by pyrene and fluoranthene. The average individual PAH concentration for all samples ranged from 0 to 25.1 ng/g milk fat; the sum of all individual PAH averages was 146.9 ng/g milk fat. The mean concentration of total PAHs in the BMI > 30 group was 224.8 ng/g milk fat, four times the PAH levels measured in women with a normal BMI. Benzo(b)fluoranthene is one of the most carcinogenic PAHs (32.08%), not found in the group of mothers with a BMI between 18.5–24.9. These results suggest that breastfed babies of obese mothers are potentially more exposed to carcinogenic PAHs [38].

In obese breastfeeding women, the duration of breastfeeding is reduced compared to breastfeeding women with a normal BMI. This could have an impact on the development of breastfed newborns [39]. In a Portuguese study, there was a relationship between age of the mother over 30 and her child’s low birth weight of less than 3 kg. Breastfed children had PAH levels of 1.41 μg/kg body weight. Unmetabolized and metabolized PAHs must be considered when calculating contamination [40]. Bariatric surgery leads to significant weight loss in the mother, which may be associated with a sustained increase in circulating lipophilic POPs in that mother. In the case of obese women of childbearing age, it is important to inform them of the risk and to quantify these potential risks for the health of the future breastfed child [41].

### 4.5. Impact of Mother’s Age

There is an interaction between age and sex, and the most significant interaction was observed for hexachlorobenzene (HCB) concentrations in serum and breast milk. For β-hexachlorocyclohexane (β-HCH) concentrations, interactions with time and pollutants were confirmed [42]. This indicates that the dynamics of pollutant excretion in breast milk are influenced by multiple factors, including the mother’s age and the specific type of pollutant. In primiparous women, HCB excretion is due to placental transfer and breastfeeding [43]. However, our analysis suggests that with increasing maternal age, the patterns of pollutant excretion, particularly HCB, may alter. In older women, this factor is no longer relevant, as discussed by Salihovic et al. [44]. The change in relevance of these factors with age underscores the complexity of understanding pollutant dynamics in breast milk and highlights the need for further research to explore how maternal age influences the transfer of pollutants to infants.

### 4.6. Toxicokinetic Modeling

There is a correlation between the concentration of contaminants in the mother’s body and the concentration of contaminants in breast milk [45]. A toxicokinetic (TK) model has been developed to quantify the amount of pollutant that has been transferred from mother to child via breast milk (LPEC) during pregnancy and lactation. The model was created to facilitate internal dosimetry calculations to assess the developmental toxicity risks associated with LPEC. We used the model to estimate whole-body concentrations in mothers and children following maternal exposures to hexachlorobenzene (HCB)N. The toxicokinetic model was tested and validated. Their analyses revealed that half-life was the most influential parameter on children’s plasma pollutant concentrations, followed by the pharmacokinetic parameters of milk/plasma partition coefficient and volume of distribution [46,47].

### 4.7. The Effect of Parity

Parity was studied as a factor that could influence dichlorodiphenyltrichloroethane (DDT) concentrations in women living in southern Mexico. The median DDE/DDT ratio was 14.7. Primiparity was one of the explanatory factors that could explain the doubling of concentrations (0.010 mg/kg and 0.868 mg/kg) in multiparous women (0.005 mg/kg and 0.583 mg/kg) (*p* < 0.05) [48]. In a Spanish study, this same parity factor was found to be correlated in POP measurements [49]. The mechanistic link between parity and POP levels in breast milk remains an area requiring further investigation. While the data suggest that parity influences the concentration of pollutants, the underlying biological mechanisms remain to be fully elucidated. This observation opens avenues for future research to explore the physiological changes during successive pregnancies that could affect the accumulation and mobilization of persistent organic pollutants.

### 4.8. What Are Their Effects on Child Health, in Particular Neurotoxicity?

The most “critical window of vulnerability” is the period of the first 1000 days of life [50]. The metabolic pathways of xenobiotics are immature, suggesting a different efficiency of metabolic and detoxification systems. The toxicity of pollutants can be explained by the very large exposure ratio in infants, given their body surface (Lorenzetti S [50]). Carcinogenicity, developmental and endocrine disruption, reproductive toxicity, and neurotoxicity are the main known serious adverse effects of POPs. Recently, Lorenzetti S (2021) observed that maternal dietary exposure to dioxins and polychlorinated biphenyls (PCBs) could be associated with language delay in Norwegian children aged 3 [50].

To study the effects of low-level Hg exposure on brain development through fish consumption and the interaction between Hg and Se, a prospective study was carried out on Italian children aged 40 months, to assess cumulative effects. A total of 900 pregnant women were included; 767 remained in the study at the time of delivery and 470 children at the age of 40 months. After excluding premature births, 456 children were analyzed. The greatest difference in terms of risk of sub-optimal neurodevelopment was observed for the category with high Total Hg and low Se, with an OR = 2.55 (90% CI 1.02; 6.41) in the multiplicative model and an OR = 1.33 (90% CI 0.80; 1.87) in the additive model. The high THg and high Se categories showed a very slight fit of the additive model (OR = 1.07, 90% CI 0.65; 1.50) compared with the multiplicative model (OR = 1.66, 90% CI 0.73; 1.77). A negative-antagonistic-interaction term for this category was estimated in the multiplicative model, giving an OR = 1.17 (90% CI 0.42; 3.28). Assessment of the effect of fish consumption on human health should also consider the various ratios between Se and Hg concentrations in different fish species [50].

Endocrine effects: BDE209, a widely used industrial brominated flame retardant (BFR), is a pollutant increasingly present in breast milk. It is a highly toxic pollutant for the thyroid gland and thyroid function and may promote thyroid cancer at multiple levels. For example, BDE209 interferes directly with thyroid, the hypothalamic–pituitary-thyroid (HPT) axis, and thyroid enzyme activities [51], which may explain its effect on neurocognitive functions. A Chinese survey studied the average estimated daily intakes (EDIs) of TBBPA, HBCD, and BDE-209 via human milk for infants aged 1–6 months were 39.2, 51.7 and 3.65 ng/kgbw/day respectively [52].

### 4.9. Can We Compare Breast and Formula Milk? Which One Is Safer?

Breast milk is the best source of nutrition for newborns, and has been shown to provide nutritional, immunological, metabolic, organic, and neurological benefits. Indeed, breast milk contains specific antibodies to protect the child, lipids beneficial for brain maturation, and a reduced risk of obesity in children [53]. For all these reasons, breast milk is superior to formula milk in terms of health benefits for the child. However, as a complex biological fluid, it consists not only of nutritional compounds but also contains environmental contaminants. From this point of view, breast milk may represent a risk. However, the real challenge lies not only in assessing the extent to which artificial milk is also contaminated but also in understanding the long-term health implications of these contaminants in both breast and formula milk.

Artificial milk comes from cow’s milk, which is not immune to POP and heavy metal contamination, given that the cow’s diet is polluted by these same contaminants. Artificial milk can also be contaminated by plastic by-products during processing, which can reach the child via this mode of feeding [54]. While the advantages of breastfeeding are well established, it has become increasingly clear that a thorough comparison of contaminant levels in both breast and formula milk is essential for providing clear, evidence-based guidance to mothers. This understanding emphasizes the need for comprehensive studies that not only compare the contaminant levels but also explore their bioavailability, metabolism, and potential health effects in infants. Such studies are crucial for advancing our understanding of the safest dietary options for infants and for developing guidelines that can help protect both mothers and infants from the adverse effects of environmental contaminants.

The absence of comprehensive comparative data on contaminant levels and their health implications in breast versus formula milk hinders our ability to fully assess the benefits and risks associated with each feeding option [55,56]. The scientific community, therefore, calls for more well-constructed research to fill this gap and provide mothers with transparent, science-based information about their choices regarding infant nutrition [57]. This effort is vital for making informed decisions that best support the health and development of the child.

## 5. Discussion

### 5.1. Can We Predict Chemical Load in Breast Milk?

Smith et al. (1987) described a method for quantifying POP contamination in breast milk. They estimated the mother’s average daily intake, an estimate of the half-life (t 1/2) of POPs as a function of body weight (BW). The Smith team’s approach calculates the biological half-life of POPs that are poorly metabolized and lipophilic. This highlights the potential for using physiologically based pharmacokinetic modeling (PBPK) as an emerging tool to better predict exposure to compounds, medicines, or pollutants in breastfeeding mothers. This concerns POPs that are bound in breast milk due to its high fat content. The biomonitoring of breastfeeding mothers through this study in the milk matrix is interesting because it’s a sensitive, reproducible method that allows comparisons over time and across different countries. Incorporating PBPK models could refine these predictions by considering individual variability and the dynamic changes in milk composition over lactation. This makes it possible to see the impact of public health policies on POPs reduction over the long term, particularly in developing and developed countries [58].

### 5.2. Some Recommendations to the Mothers and Decision-Makers

International recommendations are unanimous and concordant in justifying breastfeeding up to the child’s second birthday, with exclusive breastfeeding up to 6 months of age. Breastfeeding is the only exclusive food recommended for newborns up to 6 months of age by the WHO and EFSA [59,60]. Moreover, considering the environmental contaminants present in breast milk, it becomes crucial to pursue environmental and dietary strategies to minimize exposure. EFSA is a European scientific authority that issues recommendations based on literature analysis. It examines the question of whether foods other than breast milk or formula should be introduced, and what the consequences are for the child’s health. The need for continued research into safer dietary practices and the reduction of environmental pollutants is evident. In its opinions for breastfeeding mothers, EFSA recommends that mothers refrain from smoking tobacco, THC [61,62], refrain from using narcotics while breastfeeding, use organic food, reduce cosmetics, reduce high-risk workplaces in line with legislation, and refrain from using insecticides and pesticides at home for indoor plants. These recommendations underline the interconnectedness of environmental health and infant nutrition, further emphasizing the One Health concept.

## 6. Perspectives

POPs are characterized by their lipophilicity and bioaccumulation, which accounts for the putative presence of these pollutants in breast milk. The main objectives of this literature review were to update the origins of these pollutants and the mechanisms leading to their transfer to breast milk, the factors that may contribute to their occurrence, and their possible impact on breastfed children.

The main recommendation that can be made is to inform mothers about the benefits of breastfeeding and the risks of not breastfeeding. It is critical to always compare two conditions for benefits and risks: breastfeeding vs. not breastfeeding. The current knowledge argues in favor of breastfeeding despite the risks represented by the presence of pollutants, and this is due to the important nutritional and immunological benefits of breastfeeding. Furthermore, formula-based diets may also contain contaminants which are often poorly characterized. The impact of the duration of breastfeeding is often raised; however, very few women actually breastfeed for a long time. Parents should also be aware that children can be contaminated at different periods of their development, from the prenatal period, the in-utero period and via total oral feeding. The more informed they are, based on objective, scientific data, the better informed their choice will be. We propose to conduct a comparative scoping review exploring pesticide and other environmental exposures that formula-fed infants are exposed to.

## 7. Conclusions

This comprehensive review underscores the pervasive presence of persistent organic pollutants (POPs) and heavy metals in breast milk, highlighting the critical need for ongoing vigilance and research in this area. The inclusion of PBPK modeling in future studies could enhance our ability to predict and mitigate the risks associated with these contaminants. Our findings reveal that despite global efforts to reduce environmental contamination, pollutants such as polychlorinated biphenyls (PCBs) and dichloro-diphenyl-trichloroethane (DDT) continue to be detected in breast milk samples worldwide. The implications of these contaminants on infant health remain a significant concern, given the potential for adverse developmental and health outcomes.

Significantly, our analysis indicates that while breast milk remains the most beneficial source of nutrition for infants, the detection of these substances points to a broader environmental issue that requires comprehensive policy and regulatory action. This reaffirms the importance of adopting a One Health approach to address the multifaceted challenges of environmental contamination. The WHO’s endorsement of exclusive breastfeeding for the first six months of life underscores the importance of reducing environmental pollutant exposure to protect the health of both mothers and infants.

Future research should not only focus on longitudinal studies to assess the long-term health impacts of early-life exposure to these pollutants but also explore innovative strategies for reducing exposure through environmental and dietary interventions. Additionally, further efforts are needed to enhance the monitoring and regulation of environmental pollutants, aiming to minimize their presence in human milk. By addressing these challenges, we can safeguard the health benefits of breastfeeding and ensure a healthier start for future generations.

In light of these findings, it is imperative to continue risk reduction strategies and policies at both national and international levels to protect mothers and infants from the potential risks posed by contaminants in breast milk. Our review emphasizes the need for a collaborative effort involving policymakers, healthcare providers, environmental scientists, and the community to develop effective strategies for reducing pollutant exposure.

## Figures and Tables

**Table 1 healthcare-12-00680-t001:** Contaminants in breast milk.

Contaminant Category	Specific Contaminants	Average Concentrations	Max Allowed Levels	Health Risks
Polychlorinated biphenyls (PCBs)	Indicator PCBs	0.123 µg/g (123.12 ng/g) lipid weight [6]	0.002 µg/g bw/week (2 pg TEQ/kg bw/week, EU) [7]	Endocrine disruption, cancer, effects on nervous system development, immunotoxicity [6].
Perfluoroalkyl and polyfluoroalkyl substances (PFAS)	PFOA, PFOS	0.022 µg/L, 0.021 µg/L respectively [6]	0.07 µg/L combined (70 ppt combined, PFOA + PFOS, EPA—Water) [8]	Immune system effects, developmental delays, endocrine disruptions, fertility effects, increased risk of certain cancers [6].
Organochlorine pesticides (OCPs)	Various	Not done	0.01 mg/kg (10 µg/kg, general MRL for pesticides in EU) [9]	Neurological effects, cancer, endocrine disruptions, reproductive and developmental effects [6].
Halogenated flame retardants (HFRs)	Various	Not done	Exposure in children: 0.237–0.320 ng/kg bw/day [10]	Hormone-dependent cancers, endocrine disruptions, developmental toxicity, effects on the nervous system [6].
Dioxins (PCDDs) and furans (PCDFs)	Various	Observed decrease [11]	0.002 µg/g bw/week (2 pg TEQ/kg bw/week, EU) [7]	Cancer, reproductive and developmental problems, endocrine disruption, immune effects, chloracne [11].
Heavy metals	Lead, cadmium	Higher levels in mining and agricultural areas [12,13]	Lead: 0.020 mg/kg (Milk, EU) [10], 0.015 mg/L (15 µg/L, EPA—Water) [14]; Cadmium: 0.010 mg/kg (Infant food, EU), 0.005 mg/L (5 µg/L, EPA—Water) [15]	Neurotoxicity, hypertension, reproductive issues. Cadmium: renal toxicity, osteoporosis, increased risk of certain cancers [12].
Polycyclic aromatic hydrocarbons (PAHs)	PAHs 4 (group of four PAHs including benzo[a]pyrene)	0.0019 mg/kg in diet [16]	Benzo(a)pyrene: 0.002 mg/kg, PAHs 4: 0.010 mg/kg [16]	Cancer, genetic damage, reproductive effects, immunosuppression [16].
Dichlorodiphenyltrichloroethane (DDT)	DDT, p,p′-DDE	0.01124 mg/kg bw/day in infants [17]	0.1 mg/kg body weight/day [18]	Cancer, reproductive and developmental effects, endocrine disruptions, immunotoxicity [17].
Polybrominated diphenyl ethers (PBDEs)	BDE-153, BDE-209	Observed decrease [19] 0.00354 mg/kg of lipids [20]	0.00102 mg/kg of lipids [20]	Endocrine disruption, toxic effects on nervous system development in young children, reproductive effects [20].

## Data Availability

Data availability is not applicable to this article as no new data were created or analyzed in this study.

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
