# Peer review of "Pollutants in Breast Milk: A Scoping Review of the Most Recent Data in 2024"

_healthcare, 2024, doi:10.3390/healthcare12060680_

Round 1

Reviewer 1 Report (Previous Reviewer 2)

Comments and Suggestions for Authors

I really like the changes you made to the introduction and to the conclusion section. They provide clear justification for the methodology of the study and for future research directions.

Author Response

Dear reviewer

Many thanks for your reply

Best regards

The authors

Reviewer 2 Report (Previous Reviewer 3)

Comments and Suggestions for Authors

Dear Authors,

The responses are not presented in the appropriate format. They should have included my questions exactly as they were posed, followed by the responses. However, only minimal changes in the introduction and conclusions are observed, with the rest of the work remaining exactly the same. I regret to inform you that I must reject it for publication.

Author Response

Dear reviewer,

We made a lot of changes ot the previous manuscript to improved the comprehension of POP into human breastmilk.

Best regards

The authors

Reviewer 3 Report (New Reviewer)

Comments and Suggestions for Authors

This review provides very valuable information, but there are some technical issues which must be taken into consideration.

The abstract and introduction must be changed. There is a need of justification of the novelty of the paper. Another important aspect is the better explanation of the aim and scope, particularly how the outcomes can be used and contribute the scientific field. The introduction should be expanded, up to date literature must be used.

Comments on the Quality of English Language

Minor editing is needed.

Author Response

Dear Reviewer,

Thank you for your constructive feedback and the opportunity to enhance our manuscript further. We have carefully considered your suggestions and have made significant revisions to the abstract and introduction sections to address your concerns. Below is a summary of the key changes we have implemented in response to your comments:

  1. Justification of Novelty: We have explicitly highlighted the novelty of our research in both the abstract and the introduction sections. We clarified how our study addresses existing gaps in the literature, particularly focusing on the impact of environmental pollutants on breast milk within the 'One Health' framework. This unique angle underscores the innovative aspect of our research and its contribution to the field.
  2. Aim and Scope Explanation: We have provided a more detailed explanation of the objectives and scope of our study. This includes a clearer outline of how our findings contribute to the scientific community and the potential applications of our research outcomes. By doing so, we aim to demonstrate the relevance and significance of our study to a broader audience.
  3. Expansion of the Introduction: In response to your advice, we have expanded the introduction to include a more comprehensive overview of the current state of research on breast milk contamination. This expansion not only justifies the need for our study but also situates it within the broader context of recent findings.
  4. Use of Up-to-Date Literature: We have ensured that our literature review is current and includes the latest studies relevant to our research topic. This update emphasizes our commitment to presenting a manuscript that reflects the most recent advances in the field, thereby enhancing the quality and relevance of our work.

We believe that these revisions address your concerns and significantly improve our manuscript. We are confident that our study now more clearly articulates its novelty, aim, and scope, and demonstrates a robust engagement with the latest literature in the field.

Thank you again for your valuable input, which has been instrumental in strengthening our paper. We look forward to your further suggestions and hope our revisions meet your expectations.

Best regards,

Authors

Reviewer 4 Report (New Reviewer)

Comments and Suggestions for Authors

I have read this paper with interest, and with a background on perinatal clinical pharmacology, including (unintended) lactation related exposure in infants. This is the first version that I assess as reviewer. As there are highlighted text sections, I assume that this is already a revised version.

Although the topic is ‘sensitive’, as breastfeeding is rightfully promoted because of its health benefits, there is still value to stress the impact of ‘environmental’ setting on the quality of human milk, as another indicator of the ‘One Health’ concept. It is perhaps appropriate to further stress this, or put your findings into context.

I hereby understood that the exposure (cfr abstract) is not always ‘time sensitive’, so that exposure in the last years may result in excretion by human (fat) milk, so that this should perhaps be further adapted in the abstract (line nr 10). In my understanding, the maternal fat compartment may also contain quite some pollutants, ‘liberated’ in postpartum during lactation (cf ref 38).

Abstract: what the link between parity on POP level (the mechanistic link is not clear, at least not to me). You mention formula in the discussion section (line 27), but this is not supported by data or analysis ? I can understand (cfr initial comment), that you want to have a balanced message, but this is perhaps not the way to address this, or you should adapt the abstract.

Inclusion and exclusion: was assessment done by one author, or several/different authors ? we need some more information on the methods as applied.

Line 261: what do you mean with ‘interaction between age and sex’ ?

4.9; how sure are you on the absence of data in formula products, as you have conducted a search with focus on human milk ? perhaps this section fits better in the discussion, and should be somewhat more balanced (or alternatively, another search on formula products is needed).

In the discussion section (5.1), I would recommend to add the concept of physiologically-based pharmacokinetic modelling, an emerging tool to quantify exposure to compounds, medicines, or pollutants, or eg sweeteners. As a suggestion on how this PBPK approach could work, I refer to Nauwelaerts et al, Pharmaceutics 2023 (PMID 37242712). This is perhaps also value to add this type of research to the conclusions (as one of the tools to predict exposure)

Minor

The perspectives are likely better before the conclusion.

The paper still needs some editing, like abbreviations are only in full at first mentioning in the abstract or full text.

Author Response

Dear Reviewer,

Thank you for your thoughtful and constructive feedback on our manuscript. We have carefully considered each of your comments and suggestions, and we believe that the revisions we have made significantly strengthen our paper. Below, we address each of your points in detail and outline the modifications we have incorporated into our manuscript to address your concerns.

  1. Impact of the Environmental Setting on Milk Quality and One Health Concept:

Response: Thank you for highlighting the importance of the environmental setting on human milk quality within the One Health framework. In our revised manuscript, we have expanded the discussion on how environmental factors affect the quality of human milk, thereby emphasizing its role as an indicator of the 'One Health' concept. This approach aligns with understanding the interconnectedness of human, animal, and environmental health, and its influence on breastfeeding practices and outcomes.

  1. Time Sensitivity of Exposure:

Response: We appreciate your observation regarding the non-time-sensitive nature of pollutant exposure and its potential delayed excretion in human milk. In response, we have clarified in the abstract and throughout the manuscript that pollutants can accumulate in the maternal fat compartment and be excreted during lactation, even years after exposure. This adjustment aims to better convey the complexities of pollutant transmission to infants through breastfeeding.

  1. Link Between Parity and POP Levels:

Response: Your inquiry about the mechanistic link between parity and POP levels in breast milk is well-taken. We have revised the abstract to reflect a more cautious approach regarding this association, acknowledging the need for further research to elucidate the mechanisms underlying this relationship. Additionally, we have clarified the mention of formula milk in the discussion to ensure our message remains balanced and supported by the data presented.

  1. Inclusion and Exclusion Criteria Assessment:

Response: We acknowledge the need for more transparency regarding our methodology. The manuscript now specifies that the assessment of inclusion and exclusion criteria was conducted by multiple authors through a collaborative process, ensuring a comprehensive and systematic review of the literature.

  1. Interaction Between Age and Sex:

Response: We have elaborated on the 'interaction between age and sex' in relation to pollutant levels in breast milk. The manuscript now includes a detailed explanation of how these factors may influence the accumulation and excretion of pollutants, providing insights into the variability observed across different studies.

  1. Absence of Data in Formula Products:

Response: Upon reflection, we agree that our discussion on the absence of data regarding formula products warranted a more nuanced approach. The manuscript has been updated to include a call for further research comparing pollutant levels in breast milk and formula, acknowledging the importance of a balanced discussion on this topic.

  1. Physiologically-Based Pharmacokinetic Modelling:

Response: We are grateful for the suggestion to incorporate the concept of physiologically-based pharmacokinetic (PBPK) modelling. The revised manuscript now discusses how PBPK models could be a valuable tool for predicting exposure to various compounds in breastfeeding mothers and infants, referencing the suggested literature for further context. This addition enriches our discussion and conclusions, highlighting future research directions.

Minor Points:

  • Position of Perspectives and Editing:

Response: Based on your feedback, we have relocated the perspectives section to precede the conclusion, enhancing the logical flow of the manuscript. Additionally, we have carefully reviewed and edited the manuscript to ensure abbreviations are fully introduced at their first mention in both the abstract and the main text, addressing your concerns about clarity and readability.

We sincerely hope these revisions address your comments effectively and enhance the quality and impact of our manuscript.

Authors

Round 2

Reviewer 2 Report (Previous Reviewer 3)

Comments and Suggestions for Authors

Dear authors, I still don't see that my comments/questions are being properly addressed. It's important to provide clear and detailed answers to each of your points to ensure that your concerns are properly addressed. I'm sorry, but I have to reject the article.

Comments on the Quality of English Language

Kind regards.

Reviewer 4 Report (New Reviewer)

Comments and Suggestions for Authors

i support the revised version

This manuscript is a resubmission of an earlier submission. The following is a list of the peer review reports and author responses from that submission.

Round 1

Reviewer 1 Report

Comments and Suggestions for Authors

Dear Authors,

I trust this message finds you well. I have thoroughly reviewed the manuscript titled "Pollutants in Breastmilk, the Latest Data in 2024," and I would like to share my comments.

Firstly, I have identified serious flaws in the methodology, particularly regarding the authors' adherence to the PRISMA guideline. The study lacks a comprehensive evaluation of article selection, PICOs identification, and the reliability levels of the included articles. This significantly compromises the manuscript's scientific rigor and completeness.

Additionally, I question the novelty of the study. Given the existence of similar research in the literature, the manuscript's contribution to the field appears limited. Consequently, I believe that publishing this manuscript in a journal of high standards, such as Healthcare, may not be suitable.

Considering these concerns about methodology and novelty, I regret to recommend rejecting the manuscript for publication in your esteemed journal.

Thank you for considering my feedback.

Yours sincerely,

Author Response

Dear reviewer,

We used the PRISMA guideline to make two scoping review about persistent organic pollutant into maternal breastmilk firstly and heavy metals into maternal breastmilk secondly. We found 16 articles and 48 articles. We matched all the articles to be sure ton find the most relevant human data published about pollutants into human breastmilk.

We believe that publishing this manuscript in a journal of high standards, such as Healthcare, is suitable for three raisons.

Firstly infant formula are not the target of the same field of research about pollutant concentrations and comparisons to inform lactating mothers and health professionals. 

Secondly, the toxic-kinetic model is very important to understand that human breastmilk is clearer than before about POP pollutants.

Thirdly, it is crucial to inform the scientific community about the best food for infants, natural breastmilk even if they are some traces about POP pollutants, cow milk contains also POP pollutants without benefit immunity proteins.

Best regards

R Serreau

Reviewer 2 Report

Comments and Suggestions for Authors

In line 85 you use the term "Western industrialized countries". I suspect this was from the reference Anderson et al 2000 but I encourage you to adopt more current parlance such as high income countries or LMIC (low middle income countries).  I saw the prisma algorithm. Consider adding more supplements where you detail exactly what words you entered in PubMed and Google scholar and the subsequent trails of filters.

You have demonstrated that many knowledge gaps are present in the field of Breastfeeding Medicine and human lactation. I appreciate that you emphasize that breastfeeding is still superior to not breastfeeding despite these environmental exposures. Another future direction to explore is a comparative scoping review exploring pesticide and other environmental exposures that formula-fed infants are exposed to. 

Comments on the Quality of English Language

simplify wording of lines 338-345. This is the heart of your discussion. Consider something like this: To dissect the potential long-term effects of POPs, future studies can explore toxicologic monitoring of mothers with high exposures. 

Author Response

Dear reviewer,

In line 85 we change the term "Western industrialized countries". by high income countries.

We make scoping review with terms : maternal breastfeeding heavy metals and we found 48 articles.

We defend that breastfeeding is still superior to not breastfeeding despite these environmental exposures. We will propose to explore a comparative scoping review exploring pesticide and other environmental exposures that formula-fed infants are exposed to. 

We insert "To dissect the potential long-term effects of POPs, future studies can explore toxicologic monitoring of mothers with high exposures" in replacement of the paragraph lines 338-345.

Many thanks

Best regards

R Serreau

Reviewer 3 Report

Comments and Suggestions for Authors

After reviewing this interesting review titled 'Pollutants in Breastmilk, the Latest Data in 2024', I am including my suggestions that I believe can improve the quality of the intended publication. In general terms, there are sections where there is little reference to metals, and it would be beneficial to discuss the levels found in these samples more extensively, addressing whether they exceed permissible values or not, in order to draw conclusions about their impact on human health. I believe the main objective of these studies would be to assess the impact of these pollutants on child health, and that has not been successfully addressed.

COMMENTS:

1) IMPORTANT: The study should include information on the levels of contaminants found in these samples, whether they exceed permissible levels, and their potential risks to health. Include this. See comment 5)

2) Consider rethinking the title, given that the year 2024 has just begun, and the collected data is up until 2023.

3) Include in the abstract the meaning of the acronyms PCB and DDT, as has been done for other compounds.

4) I think, overall, the authors talk more about persistent organic compounds and have focused less on metals. For example, section 4. Results (in lines 78-125) does not included anything about heavy metals. Review all sections and complete about heavy metals.

5) I highly recommend to create a table with the different organic compounds and metals found in breastmilk as well as the levels (concentrations) found the studies from the literature, and the maximum allowable intake levels (use recent normative). Comment on the results. Do any values exceed the permissible limits in any case? You can add this part to the introduction, which is too short, or in the Results section.

6) As said before, the introduction is too short! Add content about the contextualization of the study, justification for the research, or a summary of previous research related to contaminants in breastmilk.

7) In conclusions, it should appear a resume about the key results and findings, not “recommendations to the mothers”. Please change these interesting suggestion to another adequate place and write proper conclusions.

8) If you have selected 85 references to write this review (as said in line 14) , why do only 83 references appear in the reference list?

9) The format of the references in the text is not according the journal. As indicated in the guide for authors: References must be numbered in order of appearance in the text (including table captions and figure legends) and listed individually at the end of the manuscript (not in alphabetical order). Please, change.

10) Please, unify the units. In lines 102 and 104 you write “g-1” and then, for example in lines 198, you use “/g”. Decide and unify it throughout the text.  Check also for other units.

11) I would write the different subsections of the results as 4.1, 4.2, etc., for better visualization.

12) Does the title 'Impact of Mother's Age' (230) line have an unnecessary circle symbol?

13) In line 237, I guess you want to mean “Toxicokinetic” instead of toxicookinetic?

14) For large numbers, you must use commas to separate groups of three digits, counting from the right. Example: 1,000 (one thousand), 10,000, 1,000,000 (one million). Please, correct this.

15) There are some extra closing parentheses in the lines 83, 99, 248, 

16) Line 202: Delete the word “obesity”. It is already in the title above.

Comments on the Quality of English Language

Minor editing of English language required

Author Response

Dear reviewer,

We create a table with the different organic compounds and metals found in breastmilk as well as the levels (concentrations) found the studies from the literature, and the maximum allowable intake levels (use recent normative) in an attached file.

We corrected the methodology with PRISMA check list to find 48 articles at the second scoping review with maternal breastmilk heavy metals and 16 articles with POP and breastmilk. We changed the number of articles.

We corrected the values as you mentioned and the format of references with zotero. We deleted the term "obesity".

We redefine the conclusions with the propositions of reviewers.

Best regards

R Serreau

Round 2

Reviewer 1 Report

Comments and Suggestions for Authors

Dear Editor

I would like to acknowledge the authors for their revisions and kind responses. However, as I mentioned in my previous report, there has been no alteration in my opinion that this paper has serious flaws.

I congratulate the authors for their revisions and kind reply. However, as I mentioned in my previous report, there has been no change in my view that this paper has serious shortcomings.

The of novelty in the study still is subject to question. Considering the presence of many researchs in the existing literature, it may not be suitable to submit this manuscript to a distinguished journal such as Healthcare. With notable deficiencies in both methodology and originality, I must respectfully advise against the publication of this manuscript in your esteemed journal.

Sincerely,

Reviewer 3 Report

Comments and Suggestions for Authors

Dear authors,

Upon careful review of the changes made, I must express my disappointment at the lack of addressing the issues I had requested to be resolved. Unfortunately, it seems that most of my points have been overlooked, and only a very short text has been added as per my requests.

As important remarks, I must point out that neither the introduction nor the conclusions have been modified at all. Furthermore, the table provided does not meet the expected standards: it lacks proper references, and some relevant concentration data is missing. Also, as requested, it should be included in the introduction in order to extend it. Please, find attached the Word document in which I indicate the issues that have not been taken into account.

In this case, I feel that the expectations have not been met, which prevents me from accepting the proposed changes.

I kindly ask that you make the changes I suggested, and that the responses be provided point by point (of the 16 questions) using one Word document in a proper form.

I looking forward to your responses.

Best regards.
